# Understanding the Elements of Maternal Protection from Systemic Bacterial Infections during Early Life

**DOI:** 10.3390/nu12041045

**Published:** 2020-04-10

**Authors:** Sierra A. Kleist, Kathryn A. Knoop

**Affiliations:** Department of Immunology, Mayo Clinic, Rochester, MN 55905, USA; Kleist.Sierra@mayo.edu

**Keywords:** breastmilk, late onset sepsis, bloodstream infections, enteric pathogens

## Abstract

Late-onset sepsis (LOS) and other systemic bloodstream infections are notable causes of neonatal mortality, particularly in prematurely born very low birth weight infants. Breastfeeding in early life has numerous health benefits, impacting the health of the newborn in both the short-term and in the long-term. Though the known benefits of an exclusive mother’s own milk diet in early life have been well recognized and described, it is less understood how breastfed infants enjoy a potential reduction in risk of LOS and other systemic infections. Here we review how gut residing pathogens within the intestinal microbiota of infants can cause a subset of sepsis cases and the components of breastmilk that may prevent the dissemination of pathogens from the intestine.

## 1. Introduction

Bloodstream infections (BSIs) resulting from bacterial dissemination can be extremely harmful to neonates, particularly preterm and very low birth weight (VLBW, <1500 g) newborns. Late-onset neonatal sepsis (LOS) is defined as sepsis occurring 72 h after delivery. LOS has an incidence rate of 10% in preterm infants and is associated with long-term neurological development deficiencies [1,2]. Cases resulting from bacterial BSIs account for 26% of all deaths in preterm infants. LOS will continue to be an important issue among preterm infants as there is a constant reduction of the age of viability resulting from increased medical technology for treating babies born extremely preterm and at a VLBW, those that are most at risk for neonatal BSIs [3,4].

Antibiotics are currently the first line of defense against LOS, but are possibly causing more harm than good. Empirical antibiotics are given to a majority of preterm neonates, regardless of if the infant has a positive blood culture or not, as a preemptive measure of reducing sepsis [2]. This practice may have the opposite intended effect as multiple studies [5,6] have shown an association between prolonged empirical antibiotic administration in premature babies and increased likelihood of developing LOS, necrotizing enterocolitis (NEC), and/or death. Central line placement, used for administration of parenteral nutrition and antibiotics, risks the introduction of pathogens to the bloodstream and is the likely cause of many BSIs. As such, increased hygienic practices implemented in hospitals have resulted in a stark decrease in LOS caused by normal skin commensals [7]. However, despite these efforts, LOS rates in neonates remain unchanged among cases caused by gut commensals, suggesting the bacteria are entering the bloodstream through another mechanism [7].

## 2. Breastfeeding and LOS

A preterm infant’s diet plays a crucial role in disease development or avoidance. Parenteral feedings are often the only option for delivering nutrients to VLBW infants in the days immediately following birth, but long term use is strongly associated with increased risk of LOS development [8,9]. When an infant’s organs become mature enough to handle partial or full enteral nutrition, mother’s own milk (MOM) is the preferred source of nutrition [10], though preterm infants once were formula-fed at higher rates compared to newborns delivered at term. It is logical to hypothesize ailing infants physically unable to be enterally-fed are more likely to develop LOS in connection with a frail condition [8]. When MOM is unavailable, human donor milk and/or formula are given to infants instead. A number of clinical studies have demonstrated a clear connection between feeding with MOM and protection against LOS in premature infants, in addition to the benefits of a faster transition to enteral feedings, decreased likelihood of mortality, and reduced length of hospital stay [11,12]. Further, a historical clinical observation showed an LOS incidence of 57% amongst the formula-fed infants compared to an LOS incidence of 7% in MOM-fed infants, which included partial-MOM fed infants [13]. Reduced risk of LOS was correlated with increased consumption of human milk, with the odds of LOS in a NICU cohort decreasing 19% for every 10 mL/kg dose per day of human milk [11]. While this cohort pooled infants receiving donor milk with those receiving MOM into a single human milk- fed group, more than 90% of the infants in that cohort were given MOM exclusively [11]. Recent systemic data analysis suggested a possible, though not-significant, 23% risk reduction in developing LOS among exclusively breastfed infants as compared to exclusively formula-fed infants [14]. Additional clinical observations showed similar significant results where 25% of formula-fed infants developed LOS compared to 14% of MOM-fed infants [12], supporting an initiative to promote exclusive breastfeeding as the preferred protective diet in early days of life of any enterally-fed infant. To date, use of donor milk has not shown a reduction of risk of LOS, in contrast to MOM diets [15], though mechanisms of protection unique to MOM remain unclear. More clinical data should be gathered comparing the outcomes of MOM-fed infants to those fed donor milk or fortified formula as better alternatives become available to those infants unable to be fed MOM. In extremely rare cases, LOS may be the result of contaminated breast milk [16], but in the vast majority of cases, MOM-fed infants have overall better outcomes than those who require parenteral nutrition, or other enteral diets.

## 3. Enteric Origin of Pathogens

### 3.1. Pathogens in LOS

The potential mechanism of how breastmilk may protect from bacterial BSIs and LOS initially became elucidated around 10 years ago when multiple groups observed the pathogens residing in the gastrointestinal (GI) tract prior to sepsis events [17,18,19,20]. Following birth, the gastrointestinal tract becomes colonized by commensal bacteria in a dynamic process that is initially pioneered by facultative anaerobic proteobacteria and lactobacilli [21,22,23,24,25]. Common causative pathogens of LOS can be found residing in the gut, including Gram-positive members of the lactobacillales order such as Group B *Streptococcus* (GBS) or *Enterococcus faecalis*; and Gram-negative bacilli (GNB) of the gammaproteobacteria class such as *Klebsiella pneumonia* [18], *Escherichia coli* [18], *Pseudomonas aeruginosa* [26], or *Enterobacteriaceae* species [27]. Due to the low density of the commensal flora in neonates, such pathobionts [28], bacterial species that can reside in the gut microbiota as a commensal but also have the potential to become pathogenic, can colonize the GI tract [29]. By acting as a reservoir for potential pathogens, the gut microbiota poses a risk to the neonate if it becomes dysbiotic, resulting in pathogen expansion [30,31].

Preventing enteric bacterial dissemination of pathobionts may prove to be more difficult than preventing intravenous dissemination, where increased hygienic practices have reduced LOS rates [7,32]. Additionally, animal work suggests the enteral route of infection may contribute to the virulence of sepsis pathogens when compared to the intraperitoneal route of infection [33], underlining the importance of developing therapeutics preventing enteric dissemination. Prophylactic use of oral antibiotics rarely target only potential pathogens, but also disrupts the normal developing microbiota contributing to dysbiosis in the microbial community [34,35,36]. Dysbiosis, in turn, can result in enteric infection and dissemination as pathobionts gain an increased foothold in the microbial community [30]. Antibiotic resistance is a growing concern in neonatal sepsis cases, and as such, dependence and overuse of antibiotics should be avoided [37,38].

### 3.2. Modification of the Infant Gut Microbiota

Introduction of probiotic strains of commensal bacteria has been proposed as a therapeutic strategy to treat or prevent dysbiosis of the gut microbiota and prevent a number of diseases including enteric infections and LOS [39]. Live strains of commensal bacteria such as *Lactobacillus* and *Bifidobacterium* species may improve gut health by preventing pathogen colonization and promoting the development of a healthy microbiota [40,41]. Modest improvements in gut health following single probiotic strains such as *Saccharomyces bourlardii*, *Lactobacillus reuteri*, *Lactobacillus acidophilus* and *Bifidobacterium lactis* have been observed [42], including a reduction in the time needed for progression to full enteral feeding in preterm infants given *Saccharomyces boulardii* or *B*. *lactis*, though this effect is less pronounced in exclusively formula-fed infants [43].

Probiotic mixes containing multiple strains have shown the most success in reducing the risk of LOS in enterally-fed infants [44]. Such formulations can range from a mix of three strains: *L. acidophilus*, *E. faecium* and *Bifidobacterium infantum* [45] to a mix of eight strains: *Streptococcus thermophilus*, *Bifidobacterium breve*, *Bifidobacterium longum*, *Bifidobacterium infantis*, *L. acidophilus*, *Lactobacillus plantarum*, *Lactobacillus paracasei* and *Lactobacillus delbrueckii spp bulgaricus* [46], suggesting multiple strains may have complementary roles in combination to restore intestinal health and provide protection. However, probiotics such as *Lactobacillus rhamnosus GG*, *S. boulardii*, *L. reuteri*, *Lactobacillus sporogenes*, or *B. breve* as single strains, or even mixes of multiple strains show a limited effect in reducing LOS in formula-fed infants [47]. This apparent limitation of probiotics in improving outcomes in formula-fed infants as compared to breastfed infants suggests a greater deficit in the gut health of formula-fed infants that is harder to overcome with therapeutic interventions [48,49].

The shaping of the gut microbiome by breastmilk has been repeatedly observed and formula-fed infants have a gut microbiota distinct from the community found in the GI tract of breastfed infants [50,51,52,53]. Diet during early life can influence bacterial translocation as intestinal permeability was found to be significantly decreased [54] and barrier function matured quicker in breastfed preterm infants [55] when compared to formula-fed preterm infants. In animal models, formula feeding resulted in increased bacterial translocation [56,57,58]. VLBW babies in intensive care units, those most at risk of developing sepsis, often lack access to breastmilk. Therefore, identification of protective components of breastmilk that prevent dissemination represents fertile ground for the development of therapeutics to prevent bacterial infections and LOS.

## 4. Components of Breastmilk

Breastmilk is a complex formulation of nutrients, proteins, and growth factors providing neonates with benefits beyond the incomparable nourishment. Figure 1 shows maternal protection against intestinal pathogens. Several biologically active factors promote gut health and confer protection from enteric infections to the neonate. Breastmilk changes substantially throughout lactation from colostrum in the days initially following delivery to transitional milk and then mature milk approximately two weeks following delivery. Proteins such as antibodies and growth factors are present in higher concentrations in colostrum and transitional milk compared to mature milk [59,60]. Given the biological role for such proteins, breastmilk is therefore considered one component of the “mother-breastmilk-infant triad” [61] and perhaps synchronized between mother and child to afford age-appropriate nutrition and protection to neonates provided with MOM in the first weeks of life. Such factors and proteins present in breastmilk are discussed in this section. The key components of breastmilk reviewed here all can be found in higher concentrations in colostrum and transitional milk early in lactation, asking the question that for breastmilk to have protective effects in neonates, is timing everything?

### 4.1. Antibodies

Maternal antibodies, including IgM, IgG, and IgA subtypes, offer superior protection to neonates both within the intestinal lumen, and systemically. These antibodies can provide passive immunity within the neonate to any potential systemic infections [67]. Within the lumen of the neonatal GI tract, IgA is particularly important in providing protection from invasive enteric pathogens by directly binding and preventing adherence and evasion [62,67]. Beyond protection from pathogen translocation from the intestinal lumen, maternal IgG antibodies transferred to the neonates also offered protection from E. coli within the circulation [62]. There has been modest clinical evidence that IgM administration may offer systemic protection from bacterial infections [68], though neither IgM nor IgG administration reduced LOS mortality [69]. Animal work has shown maternal antibodies educate the neonatal immune responses by dampening T cell-mediated responses in early life [70], potentially quieting inflammatory responses that may precede or accompany LOS [71,72,73]. Thus, maternal antibodies protect the neonate by preventing enteric pathogens from translocating from the intestinal lumen, and potentially limiting systemic infections and intestinal inflammation.

### 4.2. Growth Factors

Growth factors present in the breastmilk include a family of ligands that neonates can sense through the epidermal growth factor receptor (EGFR) expressed on intestinal epithelial cells [25,63,64]: epidermal growth factor (EGF), amphiregulin (AREG), heparin-binding epidermal growth factor-like factor (HB-EGF), and tumor-growth factor-alpha (TGF-α) [74]. All are found in temporal gradients, with the highest concentration in colostrum [74,75,76,77,78]. EGF is one of the most abundantly concentrated growth factors in breastmilk, though EGFR ligands perform redundant functions [79], suggesting these ligands have a necessary role in early life. EGF passes through the digestive tract resisting low pH and enzymatic degradation [80,81] and can be measured in the stool of breastfed children [64], suggesting it has a biological effect throughout the intestines. EGFR activation in the neonate results in epithelial cell division, nutrient uptake, improved intestinal barrier function, reduced bacterial translocation, and reduced Toll-like receptor signaling [63,80,81,82,83]. Recent animal work modeling decreased EGFR ligands within the GI tract of neonatal mice observed translocation of enteric pathogens resulting in a systemic infection in a model of LOS, which was reversed by oral administration of recombinant EGF [64]. Thus growth factors, through the activation of EGFR on neonatal epithelial cells can limit enteric pathogens from disseminating and potentially prevent systemic infections.

### 4.3. Lactoferrin

Lactoferrin, also present in increased concentrations in the first weeks of lactation, can remain a significant component of breastmilk for months after lactation begins [84,85,86]. The primary function of lactoferrin is to bind iron for transfer to the neonate through epithelial cell absorption, which the growing neonate utilizes as an important nutrient. Therefore, lactoferrin has anti-microbial properties primarily through the iron-binding capacity as iron sequestration can limit the amount of free iron available for bacterial growth [65,87,88]. Lactoferricin, a derivative of lactoferrin, may be directly bacteriostatic and can bind bacterial wall components, potentially limiting luminal pathogens [89,90]. Lactoferrin may also promote the development of the mucosal immune system by stimulating dendritic cells that shape intestinal immune responses and enhancing IgA production in Peyer’s patches [91]. Lactoferrin has been the target of several clinical trials as a supplement to enteral diets to protect against enteric infections and LOS [92]. While therapeutic lactoferrin may have beneficial effects on modifying the neonatal microbiome and reducing potential pathobionts [93], the efficacy of lactoferrin in reducing LOS mortality remains controversial [94,95,96]. Thus, lactoferrin protects the neonate with direct anti-microbial effects, potentially limiting pathogen colonization within the microbiota.

### 4.4. Human Milk Oligosaccharides

Oligosaccharides in breastmilk, known as human milk oligosaccharides (HMOs), support the maturation of the normal infant microbiota, which in turn provides colonization resistance to enteric pathogens [66,97,98,99,100]. These glycans are dynamically produced in the first weeks of lactation [101,102] and are essentially undigested by the infant [103], but instead utilized by the developing microbiota [52]. HMOs can be utilized as a nutrient source by commensal members of the microbiota, and also probiotic strains such as *B. breve*, as colonization of the intestinal tract of infants was associated with HMO concentration and fucosylation [104]. HMOs may also modulate the growth of potential pathogens, and have been shown to have a direct effect against the formation of GBS biofilms [105]. Additionally, observations suggest HMOs may have a direct impact on infant’s epithelial cells by increasing mucus expression [106] and promoting goblet cell maturation [107], both of which can improve barrier function by enhancing the mucus layer covering the intestinal epithelium that can prevent bacterial encroachment and translocation. Finally HMOs may have an impact on the infant’s immune system by binding c-type lectins, siglecs, galectins and selectins expressed by phagocytic and antigen-presenting cells, such as dendritic cells, monocytes and neutrophils [108,109,110]. These interactions could modulate immune responses through regulating leukocyte trafficking, influencing cytokine responses and inhibiting TLR-mediated inflammation, all of which could affect LOS development and outcomes [66,111,112]. Clinically, infants born to mothers unable to produce some forms of HMOs trended toward an increased risk of LOS [113]. Similarly, a low diversity of HMOs from mothers was associated with an increase in NEC, though was not significantly associated with an increase is LOS cases [114]. These clinical data suggest more work needs to be compiled regarding supplementation of infant diets with HMOs to potentially improve gut function and sepsis outcomes [115]. Thus, HMOs may protect the neonate by shaping the microbiota, which may protect from pathogen colonization.

## 5. Future Directions

### 5.1. Supplements and MOM Alternatives

Donor milk represents a worthy alternative when MOM is unavailable, though reports have shown the processing of donor milk, including pasteurization and potentially multiple freeze-thaw cycles, may reduce the concentrations of the beneficial proteins, particularly immunoglobulins and lactoferrin [74,116]. Pasteurization of MOM showed a non-significant trend of increased infectious LOS morbidity [117], suggesting the reduction of these beneficial components in MOM could lead to an increased risk of LOS. Pasteurization is an imperative step in donor milk processing to prevent potential transmission of pathogens through contaminated breast milk [16], and work optimizing pasteurization processes to remove pathogenic threat while maintaining beneficial factors is currently being completed [118,119]. The Holder pasteurization method, the recommended pasteurization method of donor milk, sterilizes bacteria present in the milk, and does not degrade growth factors such as EGF and TGF-β [119]. Immunoglobulins and lactoferrin are reduced following Holder pasteurization, though may be protected following high temperature short time (HTST) treatments, an experimental pasteurization method [120,121]. Additionally, the increased concentrations of these proteins early in lactation suggests age-matched donor milk or supplements containing a combination of immunoglobulins, growth factors, lactoferrin, and HMOs at concentrations found in colostrum may represent the next steps in the progression toward an appropriate alternative when MOM cannot be provided.

### 5.2. Animal Modeling

While clinical interventions are quickly being brought to the NICU, animal models are revealing potential mechanisms of acquisition and protection against LOS. Therefore, the development and use of animal models that represent how LOS is clinically acquired is essential [30,62,64]. Traditional models of LOS have been limited to intravenous injection of cecal contents or single bacterial components, such as lipopolysaccharide. While these models may help elucidate how neonates respond to systemic bacterial insults, these models lack insight into the enteric route of entry pathogens may utilize. Injection of cecal contents may introduce too many bacterial elements as most LOS patients are infected with only one bacterial species at a time. Similarly, intravenous injection of lipopolysaccharide, a component of some bacterial cell walls, may reduce relevance as there can be strain variation in LOS pathology within the same bacterial species [64]. As increased hygienic practices reduce the number of intravenous-acquired LOS cases, similar interest should be placed on reducing the number of LOS cases resulting from an enteric route of origin, with animal work modeling the oral route of pathogen entry. Understanding how MOM and the developing microbiota protect the neonate from enteric pathogens will provide clear directions for future therapeutics.

## 6. Conclusions

While the clinical connection between breastfeeding and reduced LOS risk is currently a potential, though logical, correlation, the many components within breastmilk offer observable benefits to the developing neonate, particularly within the intestinal environment. If enteric pathogens continue to threaten infants and cause a substantial portion of LOS cases, factors present in breastmilk may provide exceptional protection to the neonate, representing strong candidates for supplementation of breast milk that could prevent of translocating pathogens. Clinical measures such as the reduction of unnecessary antibiotics to protect the intestinal microbiome, promotion of an exclusive MOM diet when available, and sophistication of supplements combining immunoglobulins, growth factors, lactoferrin, and HMOs to formula or donor milk could result in further reduction of LOS cases by focusing on protection from enteric pathogens.

## Figures and Tables

**Figure 1 nutrients-12-01045-f001:**
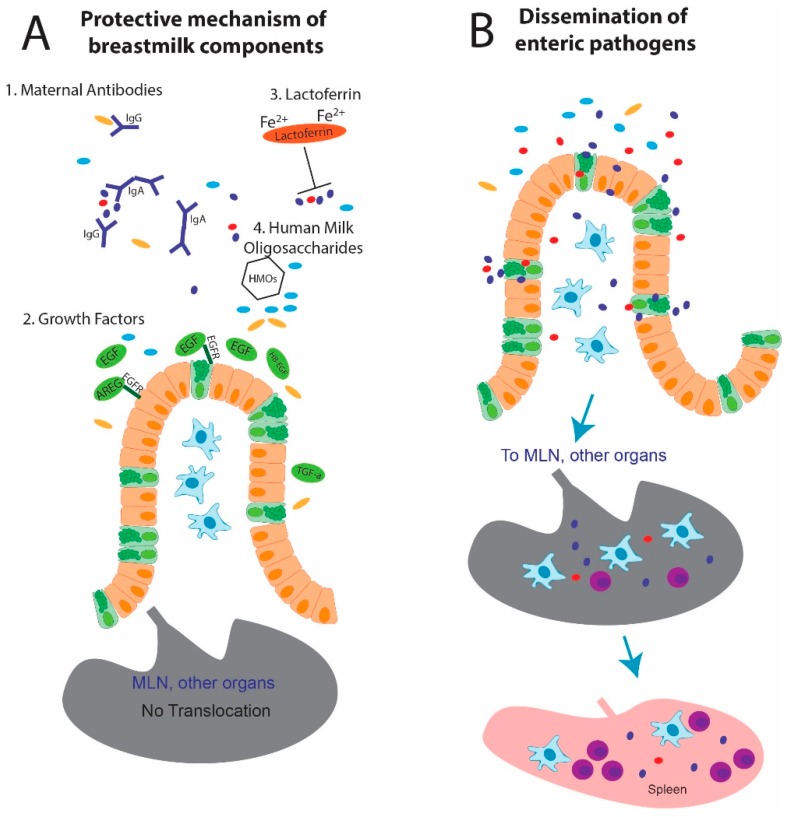
Maternal protection from enteric pathogens. (**A**) Components in breastmilk can limit enteric pathogen dissemination. (1) Maternal antibodies (IgG, IgA) can bind bacteria and directly inhibit pathogen adherence and invasion [62]. (2) Growth factors [epidermal growth factor (EGF), amphiregulin (AREG), heparin-binding epidermal growth factor-like factor (HB-EGF), and tumor-growth factor-alpha (TGF-α)] bind the epidermal growth factor receptor (EGFR) on epithelial cells to promote barrier function by cell proliferation and growth [63], and by limiting translocation via goblet cells [64]. (3) Lactoferrin sequesters iron which limits pathogen growth [65]. (4) Human milk oligosaccharides (HMOs) promote the development of the intestinal microbiota [66], which can offer colonization resistance to enteric pathogens [30]. (**B**) In the absence of these factors, pathogens can colonize the intestine lumen, cross the epithelium potentially through goblet cells [64], and disseminate to organs through the system, including the mesenteric lymph node (MLN) and spleen, resulting in late-onset sepsis (LOS).

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
