# Peer review of "Understanding the Elements of Maternal Protection from Systemic Bacterial Infections during Early Life"

_nutrients, 2020, doi:10.3390/nu12041045_

Round 1
Reviewer 1 Report
The manuscript by Kleist at al. review how gut residing pathogens within the intestinal microbiota of infants can cause a subset of sepsis cases and the components of breastmilk that may prevent the dissemination of pathogens from the intestine. It is a very important field in neonatology research and the authors give an interesting angle to it.
Major comments:
- The conclusion (and the title) is too strong. The authors hypothesize that the the age of breast milk ("timing") is crucial for its preventive effects. It is ok to discuss it speculatively in the the text, but evidence for it is too weak to have it in the conclusion (change both in title, abstract and in the end of the text)
- It would be better to conclude that there are candidates for supplementation of breast milk that could reduce LOS originating from the gut (such as specific HMOs, growth factors)
- Mention important factors (and not only references) that are reduced by pasteurization. They could be candidates for future supplements of donor milk.
- Suggestion: include a separate paragraph about future candidates for treatment: for examples components that are reduced by pasteurization and HMOs such as DSLNT (Autran et al Gut. 2018 Jun;67(6):1064-10709. Thus discuss animal models separately.
Minor commets:
- Are there references supporting a weaker effect of donor milk, as compared to mothers own milk, on LOS? Include them.
- Add recent mechanistic papers showing how HMO may increase the barrier function (for example Wu et al. Mol Nutr Food Res. 2019 Feb;63(3):e1800658. doi: 10.1002/mnfr.201800658. Epub 2018 Dec10)
- Despite being associated to NEC incidence, HMOs have not been linked to sepsis reduction in clinical trials in preterm infants. Add the recent paper i Nutrients by Wejryd et al (Nutrients. 2018 Oct 20;10(10). pii: nu10101556. doi: 10.3390/nu10101556.) showing an association between HMO diversity with NEC but not sepsis in extremely preterm infants.
Author Response
|
Major comments:
|
Author’s response |
|
1. The conclusion (and the title) is too strong. The authors hypothesize that the age of breast milk ("timing") is crucial for its preventive effects. It is ok to discuss it speculatively in the text, but evidence for it is too weak to have it in the conclusion (change both in title, abstract and in the end of the text).
|
We agree with the reviewer regarding the current lack of data to substantiate the timing or age of breast milk is crucial for its preventative effects. Therefore we have changed the title to “Understanding the Elements of Maternal Protection from Systemic Bacterial Infections During Early Life” (Line 2-4). We have revised the abstract (Line 9-16) and revised the conclusion to read “factors present in breastmilk may provide exceptional protection to the neonate, representing strong candidates for supplementation of breast milk that could prevent of translocating pathogens.” (Line 275-277). Additionally, the phrase “synchronized use of donor milk” (line 279) has been omitted. |
|
2. It would be better to conclude that there are candidates for supplementation of breast milk that could reduce LOS originating from the gut (such as specific HMOs, growth factors).
|
We agree and have revised the conclusion to reflect this view: Line 275-276 now reads “factors present in breastmilk may provide exceptional protection to the neonate, representing strong candidates for supplementation of breast milk that could prevent of translocating pathogens” and Line 279-280 now reads “sophistication of supplements combining immunoglobulins, growth factors, lactoferrin, and HMOs to formula or donor milk could result in further reduction of LOS cases by focusing on protection from enteric pathogens.” |
|
3. Mention important factors (and not only references) that are reduced by pasteurization. They could be candidates for future supplements of donor milk.
|
We agree and have expanded the paragraph 5.1 to specify that immunoglobulins and lactoferrin are most at risk to be reduced by pasteurization (line 240), including Holder pasteurization, the current recommended pasteurization method for donor milk. We have included the following sentences regarding important factors reduced by pasteurization: (Line 246-249) The Holder pasteurization method, the recommended pasteurization method of donor milk, sterilizes bacteria present in the milk, and does not degrade growth factors such as EGF and TGF-β [120]. Immunoglobulins and lactoferrin are reduced following Holder pasteurization, though may be protected following high temperature short time (HTST) treatments, an experimental pasteurization method [121,122]. |
|
4. Suggestion: include a separate paragraph about future candidates for treatment: for examples components that are reduced by pasteurization and HMOs such as DSLNT (Autran et al Gut. 2018 Jun;67(6):1064-10709. Thus discuss animal models separately. |
We agree and have split section 5, Future Directions, (line 235) into two sections: 5.1 Supplements and MOM alternatives (Line 236) and 5.2 Animal Modeling (Line 254) |
|
Minor comments |
Author’s response |
|
1. Are there references supporting a weaker effect of donor milk, as compared to mothers own milk, on LOS? Include them.
|
To our knowledge, no studies have been completed directly comparing donor milk to Mom’s own milk (MOM) for the outcome of LOS. Studies assessing donor milk have not shown DM reduces risk of LOS; Meier, P et al, Donor Human Milk Update: Evidence, Mechanisms, and Priorities for Research and Practice. The Journal of pediatrics 2017, 180, 15-21, doi:10.1016/j.jpeds.2016.09.027.
We have included this reference and the following text (Line 74): “To date, use of donor milk has not shown a reduction of risk of LOS, in contrast to MOM diets [15], though mechanisms of protection unique to MOM remain unclear.”
We have also delineated which studies included only MOM-fed infants and which studies included donor-milk fed infants in a pooled “human milk” category. Line 65-69 now reads: “Reduced risk of LOS was correlated with increased consumption of human milk, with the odds of LOS in a NICU cohort decreasing 19% for every 10 ml/kg dose per day of human milk [11]. While this cohort pooled infants receiving donor milk with those receiving MOM into a single human milk- fed group, more than 90% of the infants in that cohort were given MOM exclusively [11].” |
|
2. Add recent mechanistic papers showing how HMO may increase the barrier function (for example Wu et al. Mol Nutr Food Res. 2019 Feb;63(3):e1800658. doi: 10.1002/mnfr.201800658. Epub 2018 Dec10 )
|
We agree with the reviewer that the role of HMOs in improving the intestinal barrier of infants is highly pertinent to our review and have included the suggested citation, along with the citation (Cheng, L et al, Molecular nutrition & food research 2020, 64, doi:10.1002/mnfr.201900976.) in the following sentence (Line 220-223) “Additionally, observations suggest HMOs may have a direct impact on infant’s epithelial cells by increasing mucus expression [113] and promoting goblet cell maturation [114], both of which can improve barrier function by enhancing the mucus layer covering the intestinal epithelium that can prevent bacterial encroachment and translocation.” |
|
3. Despite being associated to NEC incidence, HMOs have not been linked to sepsis reduction in clinical trials in preterm infants. Add the recent paper i Nutrients by Wejryd et al (Nutrients. 2018 Oct 20;10(10). pii: nu10101556. doi: 10.3390/nu10101556.) showing an association between HMO diversity with NEC but not sepsis in extremely preterm infants .
|
We agree and have included this citation in the following sentence (Line 230-231) “Similarly, a low diversity of HMOs from mothers was associated with an increase in NEC, though was not significantly associated with an increase is LOS cases [115].” |
Reviewer 2 Report
The paper analysis current knowledge of the potential role of breastmilk components and intestinal microbiome profile in reducing the sepsis rate in neonates
The subject of the review is very well defined and presented. All the definitions used in the paper are clearly described.
The review is properly structured, nicely written and well documented with current bibliography.
I would like the authors to address minor comments:
- Page 2 line 53-55 and 64-66 Could you comment on the differences in LOS, mortality and hospital stay between neonates fed MOM or donor milk? Do we have any data so far?
- page 3 line 99-103 As we know that the probiotocs activity is strain specific I would suggest to specify the strains that are reffered, bibliography position 42,43,44
The review is of clinical significnace and also may have practical implications.
Author Response
|
minor comments:
|
Author’s response: |
|
1. Page 2 line 53-55 and 64-66 Could you comment on the differences in LOS, mortality and hospital stay between neonates fed MOM or donor milk? Do we have any data so far?
|
See also, response to Reviewer 1 minor comment 1.
Data focusing on donor milk fed infants is currently minimal. Data suggests donor milk can reduce the risk of NEC similar to MOM, but to our knowledge, no studies have been completed comparing donor milk directly to Mom’s own milk (MOM) in respect to LOS outcomes.
Studies assessing donor milk have not shown DM reduces risk of LOS; (Meier, P et al, Donor Human Milk Update: Evidence, Mechanisms, and Priorities for Research and Practice. The Journal of pediatrics 2017, 180, 15-21, doi:10.1016/j.jpeds.2016.09.027.)
We have included this reference and the following text (line 74): “To date, use of donor milk has not shown a reduction of risk of LOS, in contrast to MOM diets [15], though mechanisms of protection unique to MOM remain unclear.”
We have also delineated which studies included only MOM-fed infants and which studies included donor-milk fed infants in a pooled “human milk” category. Line 65-69 now reads: “Reduced risk of LOS was correlated with increased consumption of human milk, with the odds of LOS in a NICU cohort decreasing 19% for every 10 ml/kg dose per day of human milk [11]. While this cohort pooled infants receiving donor milk with those receiving MOM into a single human milk- fed group, more than 90% of the infants in that cohort were given MOM exclusively [11].” |
|
2. page 3 line 99-103 As we know that the probiotocs activity is strain specific I would suggest to specify the strains that are reffered, bibliography position 42,43,44
|
We agree, as probiotic strains function with unique activity. We have revised the probiotic section to include strain specificity, and define probiotic mixes, when available. Lines 111-125 now reads: “Modest improvements in gut health following single probiotic strains such as Saccharomyces bourlardii, Lactobacillus reuteri, Lactobacillus acidophilus and Bifidobacterium lactis have been observed [42], including a reduction in the time needed for progression to full enteral feeding in preterm infants given Saccharomyces boulardii or Bifidobacterium lactis, though this effect is less pronounced in exclusively formula-fed infants [43].
Probiotic mixes containing multiple strains have shown the most success in reducing the risk of LOS in enterally-fed infants [44], probiotics[44]. Such formulations can range from a mix of three strains: Lactobacillus acidophilus, Enterococcus faecium and Bifidobacterium infantum [45] to a mix of eight strains: Streptococcus thermophilus, Bifidobacterium breve, Bifidobacterium longum, Bifidobacterium infantis, Lactobacillus acidophilus, Lactobacillus plantarum, Lactobacillus paracasei and Lactobacillus delbrueckii spp bulgaricus [46], suggesting multiple strains may have complementary roles in combination to restore intestinal health and provide protection. However, probiotics such as Lactobacillus rhamnosus GG, Saccharomyces bourlardii, Lactobacillus reuteri, Lactobacillus sporogenes, or Bifidobacterium breve as single strains, or even mixes of multiple strains show a limited effect in reducing LOS in formula-fed infants [47].”
|
Reviewer 3 Report
1. Introduction: I would recommend deleting the first paragraph and start the review with the second paragraph to better focus the reader to the question at hand 2. The subheading " enteric origins of pathogens" should be further subdivided into Pathogens in LOS and Modification of enteric pathogens to reduce LOS, to reduce the confusion associated with the switch in focus within this section 3. What is the exact incidence of LOS in exclusively breastfed babies compared to formula fed infants? 4. Does increasing Bifidobacteria via HMOs reduce LOS? Please address specifically. 5. Is there a correlation between intestinal butyrate and LOS? Please elaborateAuthor Response
|
Comments and Suggestions for Authors
|
Author’s response: |
|
1. Introduction: I would recommend deleting the first paragraph and start the review with the second paragraph to better focus the reader to the question at hand |
We agree and have removed the paragraph. One sentence, “Breastfeeding in early life has numerous health benefits, impacting the health of the newborn in both the short-term and in the long-term.” was moved to the abstract (Line 10) and one phrase “very low birth weight (VLBW, <1500 grams)” was moved to Line 30 to define the term VLBW. |
|
2. The subheading " enteric origins of pathogens" should be further subdivided into Pathogens in LOS and Modification of enteric pathogens to reduce LOS, to reduce the confusion associated with the switch in focus within this section |
We agree and have subdivided section 3 Enteric Origin of Pathogens (line 81) into 3.1 Pathogens in LOS (Line 81) and 3.2 Modification of the infant gut microbiota (Line 106) to better focus the reader’s attention to the use of probiotics to manipulate the microbiota. |
|
3. What is the exact incidence of LOS in exclusively breastfed babies compared to formula fed infants? |
We agree that including the exact incidences of LOS in MOM-fed infants to formula-fed infants is important to understand how protective breastmilk can be. We have included exact incidences from cited clinical studies and systemic meta-analysis in Line 63-73
“Further, a historical clinical observation showed an LOS incidence of 57% amongst the formula-fed infants compared to an LOS incidence of 7% in MOM-fed infants, which included partial-MOM fed infants [13]. Reduced risk of LOS was correlated with increased consumption of human milk, with the odds of LOS in a NICU cohort decreasing 19% for every 10 ml/kg dose per day of human milk [11]. While this cohort pooled infants receiving donor milk with those receiving MOM into a single human milk- fed group, more than 90% of the infants in that cohort were given MOM exclusively [11]. Recent systemic data analysis suggested a possible, though not-significant, 23% risk reduction in developing LOS among exclusively breastfed infants as compared to exclusively formula-fed infants [14]. Additional clinical observations showed similar significant results where 25% of formula-fed infants developed LOS compared to 14% of MOM-fed infants [12]” |
|
4. Does increasing Bifidobacteria via HMOs reduce LOS? Please address specifically. |
We do not know of work done examining how Bifidobacteria increase following HMO administration in infants and if such an expansion reduces LOS. We do now include the citation which shows Bifidobacterium breve colonization depends on HMO quality and quantity. Line 216-218 now reads “HMOs can be utilized as a nutrient source by commensal members of the microbiota, and also probiotic strains such as Bifidobacterium breve, as colonization of the intestinal tract of infants was associated with HMO concentration and fucosylation [112].“
Bifidobacteria alone administered as a probiotic does not reduce risk of LOS, though in a probiotic mix, can reduce risk of LOS amongst human milk fed infants, but not formula fed infants. (Line 117-125) Additionally, HMOs as produced by mothers, has only been casually associated with LOS risk and not shown to be statistically significant. (Line 229-231) |
|
5. Is there a correlation between intestinal butyrate and LOS? Please elaborate. |
We do not know of work examining intestinal butyrate concentrations and LOS. Short chain fatty acids can be found in breast milk and also found in the intestinal lumen when produced by the microbiota. Short chain fatty acids, including butyrate are known to have a immunomodulatory effect on the host, though to our knowledge, minimal work has been completed examining butyrate in neonatal sepsis.
Singer et al (reference 30) found no correlation between butyrate and LOS, in an animal model. However, as we have not focused on short chain fatty acids in this review and we have therefore not mentioned this observation. |
Round 2
Reviewer 1 Report
The revision has improved the manuscript.